# INTERPRETABLE RELATIONAL REPRESENTATIONS FOR FOOD INGREDIENT RECOMMENDATION SYSTEMS

## ABSTRACT

Supporting chefs with ingredient recommender systems to create new recipes is challenging, as good ingredient combinations depend on many factors like taste, smell, cuisine style, texture among others. There have been few attempts to address these issues using machine learning. Useful Machine Learning models do obviously need to be accurate but importantly – especially for food professionals – interpretable. In order to address these issues, we propose the Interpretable Relational Representation Model (IRRM). The main component of the model is a key-value memory network to represent relationships of ingredients. We propose and test two variants of the model. One can learn latent relational representations over a trainable memory network (Implicit model), and the other can learn explainable relational representations over a pre-trained memory network that integrates an external knowledge base (Explicit model). The relational representations resulting from the model are interpretable – they allow to inspect why certain ingredient pairings have been suggested. The Explicit model additionally allows to integrate any number of manually specified constraints. We conduct experiments on two recipe datasets, including CulinaryDB with 45,772 recipes and Flavornet with 55,001 recipes, respectively. The experimental results show that our models are both predictive and informative.

## 1 INTRODUCTION

Data mining and machine learning methods play an increasingly prominent role in food preference modeling, *food ingredient pairing discovery* and *new recipe generation*. Solving these tasks is non-trivial, since the goodness of ingredient combinations depends on many factors like taste, smell, cuisine, texture, and culture. Ahn et al. (2011) detected that the number of shared flavor molecules between ingredients is one of important factors for food pairing. They found Western cuisines show a tendency to use ingredient pairs that share many flavor compounds, while East Asian cuisines tend to avoid compound sharing ingredients. Using this idea, Garg et al. (2017) developed a rule-based food pairing system which ranks ingredients based on the number of shares of flavor molecules. Recently, Park et al. (2019) suggested a neural network approach based on flavor molecules and co-occurrence of ingredients in recipes. These approaches focus on one-to-one food pairing. There is also research related to many-to-one pairing. De Clercq et al. (2016) proposed the *Recipe Completion Task* which tries to identify matching ingredients for a partial list of ingredients (the recipe) using a Matrix Factorization based recommender system. Although efforts have been made to detect good ingredient combinations, there is no current Machine Learning method in this field that allows to interpret why suggested pairs are good.

Our work is targeted at interpretable recommendation systems for food pairing and recipe completion. Given a set of pre-selected ingredients (cardinality 1 or more) by a user, the recommender suggests top-N ingredients from a set of candidates. For example, suppose a user selects *apple* and *chocolate* as the pre-selected ingredients, our recommender suggests some good paired ingredients (e.g. *cinnamon*) and also identifies reasons (e.g. *cinnamon* is good for *apple* and *chocolate* in terms of their flavor affinity).

For this, we propose the Interpretable Relational Representations Model (IRRM) in two variants to address food pairing and recipe completion tasks. The model features a key-value memory network (Sukhbaatar et al. (2015), Miller et al. (2016)) to represent relationships of ingredients. One variant

of the model is trained to learn latent relational representations over a trainable memory network (Implicit Model). The other model can learn explainable relational representations over the pre-trained memory network integrating an external knowledge base (Explicit Model). The relational representations are interpretable and can be queried as to the reasons why the ingredients have been suggested. The Explicit model can integrate any number of constraints which can be decided manually based on the characteristics of the desired recommender system. Our contributions are as follows:

1. We model ingredient pairing as a general recommendation task with implicit feedback.

2. We introduce the Interpretable Relational Representations Model and it's two variants: Implicit and Explicit. Both of which can learn pair specific relational representations (vectors) for one-to-one (i.e. ingredient to ingredient) and many-to-one (ingredient-set to ingredient) food pairing tasks. The relational vectors are also interpretable.

3. We propose a training procedure to learn one-to-one and many-to-one relationships effectively using recipes.

4. We evaluate our proposed models in the Recipe Completion Task and the Artificial Food pairing Task on the CulinaryDB and the Flavornet datasets. Our proposed approaches demonstrate competitive results on all datasets, outperforming many other baselines.

5. We perform qualitative analysis. The results presents our proposed Explicit model is capable of unraveling hidden ingredients structures within recipes.

## 2 RELATED WORK

There are two related streams of work in recommender systems that are important for this paper: the *session-based setting* and the *knowledge-aware systems*.

In the session-based setting, user profile can be constructed from past user behavior. A natural solution to this problem is the item-to-item recommendation approach.A variety of methods exist for this problem. For example, Quadrana et al. (2017) models the item sequence using RNNs, Kang & McAuley (2018) uses Self-Attention layers, and Wu et al. (2020) uses Transformer layers. While these methods mainly focus on how to encode item click-sequence interactions, we target good ingredient pairing using only ingredient attributes and the relationship between a ingredient set and an ingredient based on co-occurrence in recipes. For this we develop a new architecture integrating set encoders and relational memory with novel loss and score functions.

There are also increasingly methods for integrating knowledge into recommenders. Zhang et al. (2016) and Cheng et al. (2016) directly incorporate user and item features as user profile into neural network models. Huang et al. (2018) and Wang & Cai (2020) integrate them using a pre-trained knowledge graph. These methods try to represent user context using external knowledge base, therefore, usually these knowledge embeddings are integrated to user embeddings. In this work, we incorporate knowledge specifically to detect relationships between an ingredient set and an ingredient for interpretation to improve recommendation performance.

## 3 PROBLEM DEFINITION

We first introduce the notations used throughout this paper. We model recipe completion as a recommendation scenario with implicit feedback (Huang et al., 2018, Tay et al., 2018. In such scenarios, a user has interacted with an item and the system infers the item that user will interact next based on the interaction records of the user. We apply this to the food domain by using recipes as interaction records.

Let $\mathcal{I}$ denote a set of ingredients and $\{i_1, \ldots, i_M\}$ denote a pre-selected ingredient set, where $i \in \mathcal{I}$ is the ingredient and $M$ is the number of ingredients. We call $\{i_1, \ldots, i_M\}$ pre-selected ingredient set in this paper. Next, let $\mathcal{I}_{candidate}$ denotes a set of candidate ingredients. $\mathcal{I}_{candidate}$ depends on each pre-selected ingredient set, that is, $\mathcal{I}_{candidate} = \mathcal{I} - \{i_1, \ldots, i_M\}$. In addition, we assume that a knowledge base (KB) of ingredients is also available and the KB contains factors which are related to why some ingredients are good combinations. A KB is defined as a set of triplets over a

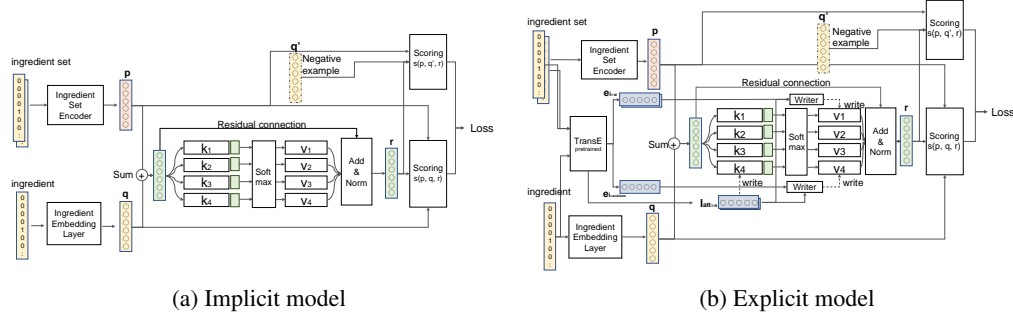

(a) Implicit model           (b) Explicit model

Figure 1: IRRM architectures

entity set $\mathcal{E}$ and a relationship set $\mathcal{L}$. A KB triplet $\langle e_i, l, e_a \rangle$ composed of two entities $e_i, e_a \in \mathcal{E}$ and a relationship $l \in \mathcal{L}$, where $e_i$ is an ingredient (e.i. $e_i \in \mathcal{I}$) and $l$ is an attribute and $e_a$ is the attribute value. For instance, $\langle$*apple, flavorMolecule, (-)-Epicatechin*$\rangle$ denotes that *apple* contains the *(-)-Epicatechin* flavor molecule.

Based on these preliminaries, we define the food ingredient recommendation task. Given a pre-selected ingredient set $\{i_1, \ldots, i_M\}$ and candidate ingredients $\mathcal{I}_{candidate}$, we would like to infer the top-N ingredients from $\mathcal{I}_{candidate}$.

## 4 RECOMMENDATIONS WITH MEMORY NETWORKS

In this section, we introduce the IRRM architectures. We start with the Implicit model that consists of a trainable key-value memory network. We then augment the Implicit model using a key-value memory network which integrates pre-trained entity and relationship vectors with ingredient attributes in the KBs – we call this extention the Explicit model. The overall architecture is described in Figure 1. The input of our architecture are a pre-selected ingredient set and a candidate ingredient $i_{candidate} \in \mathcal{I}_{candidate}$. The output is a score. In inference, our recommender uses these scores to rank $\mathcal{I}_{candidate}$.

### 4.1 INGREDIENT EMBEDDING LAYER AND INGREDIENT SET ENCODER

Ingredients are represented as one-hot encoding vectors (corresponding to a unique index key belonging to each ingredient). At the embedding layer, this one-hot encoded vector is converted into a low-dimensional real-valued dense vector representation which is multiplied with the embedding matrices $\boldsymbol{Q} \in \mathbb{R}^{d \times |\mathcal{I}|}$ – which stores ingredient embeddings. $d$ is the dimensionality of the ingredient embeddings while $|\mathcal{I}|$ is the total number of ingredients. $i_{candidate}$ is converted to $\boldsymbol{q}$ using this embedding layer. On the other hand, pre-selected ingredients $\{i_1, \ldots, i_M\}$ are encoded by the Ingredient Set Encoder (Figure 6). At first, each ingredient $i_j$ is converted to a vector using the Ingredient Embedding Layer (same as $i_{candidate}$). As a result, $\boldsymbol{i}_j \in \mathbb{R}^d$ vectors are generated. The sum of these vectors is converted to the ingredient set vector $\boldsymbol{p}$ using a feed-forward network with a single hidden layer, followed by Layer Normalization.

### 4.2 RELATION ENCODER

Tay et al. (2018) introduced LRAM (Latent Relational Attentive Memory), in order, to generate latent relational vectors between user-item interactions. We expand this module by adding a residual connection, followed by Layer Normalization. This idea is inspired by Vaswani et al. (2017).

Given the pair of a pre-selected ingredient set vector and a candidate ingredient vector, $\langle \boldsymbol{p}, \boldsymbol{q} \rangle$, the Relation Encoder first applies $\boldsymbol{s} = \boldsymbol{p} + \boldsymbol{q}$ to generate the joint embedding of $\boldsymbol{p}$ and $\boldsymbol{q}$. The generated vector $\boldsymbol{s} \in \mathbb{R}^d$ is of the same dimension of $\boldsymbol{p}$ and $\boldsymbol{q}$. Note we also tried other transfer functions here such as element-wise multiplication or just using a multi-layered perceptron $MLP(\boldsymbol{p}, \boldsymbol{q})$. However, we found that addition performs best. This joint embedding $\boldsymbol{s}$ is used as the input of the memory network. The attention vector $\boldsymbol{a} \in \mathbb{R}^d$ is a vector of importance weights over keys which are

represented as the key matrix $\boldsymbol{K} = [\boldsymbol{k}_1, \ldots, \boldsymbol{k}_N]^T \in \mathbb{R}^{N \times d}$, where $N$ is the number of key-value pairs in the memory network and $\boldsymbol{k}_j \in \mathbb{R}^d$ is a key vector. Each element of the attention vector $\boldsymbol{a}$ can be defined as $a_j = \boldsymbol{s}^T \boldsymbol{k}_j$, where $a_j \in \mathbb{R}$. In order to normalize the attention vector $\boldsymbol{a}$ to a probability distribution, we use the Softmax function: $Softmax(a_j) = \frac{\exp(a_j)}{\sum_{n=1}^{N} \exp(a_n)}$. We generate the vector $\boldsymbol{m} = \sum_{n=1}^{N} Softmax(a_n) \boldsymbol{v}_n$ as the summation of weighted value vectors which are represented as the value matrix $\boldsymbol{V} = [\boldsymbol{v}_1, \ldots, \boldsymbol{v}_N]^T \in \mathbb{R}^{N \times d}$. Finally, in order to generate the relational vector $– \boldsymbol{r}$, $\boldsymbol{m}$ is added with the joint embedding $\boldsymbol{s}$ and Layer Normalization is applied as follows $\boldsymbol{r} = LayerNorm(\boldsymbol{s} + \boldsymbol{m})$.

### 4.2.1 THE EXPLICIT MODEL

In order to improve interpretability and predictive performance, we incorporate ingredient attribute information from a given KB into the memory network. Inspired by recent works which integrate a memory network with external memories (Huang et al. (2018)), we propose the Explicit Relational Encoder. Instead of the trainable key matrix $\boldsymbol{K}$ and value matrix $\boldsymbol{V}$, we pre-train vectors over a given KB. We then freeze key and value matrix for training the explicit model. Given a pair of a pre-selected ingredient set $\{i_1, \ldots, i_M\}$ and a candidate ingredient $i_{candidate}$, $\{i_1, \ldots, i_M, i_{candidate}\}$ is converted into the entity vectors using the KB embeddings which provide the entity vectors $\boldsymbol{e} \in \mathbb{R}^{d^{KB}}$ and the relationship vectors $\boldsymbol{l} \in \mathbb{R}^{d^{KB}}$. Note that in case of $d^{KB} \neq d$, we convert the joint embedding $\boldsymbol{s} \in \mathbb{R}^d$ into $\boldsymbol{s}' \in \mathbb{R}^{d^{KB}}$ and the relational vector $\boldsymbol{r} \in \mathbb{R}^{d^{KB}}$ into $\boldsymbol{r}' \in \mathbb{R}^d$ with linear projections. We use the TransE (Bordes et al. (2013)) for the KB embeddings. The reason for this choice is that given triplet $\langle e_i, l_{att}, e_{att}^i \rangle$, TransE can learn entity vectors and relationship vectors to follow $\boldsymbol{e}_{att}^i = \boldsymbol{e}_i + \boldsymbol{l}_{att}$. KB relationships usually correspond to attribute types of entities, so we use the notation $l_{att}$ as the attribute type and $e_{att}^i$ as its value. Hence, we set the key matrix as follows:

$$\boldsymbol{K} = [\boldsymbol{l}_{att_1}, \ldots, \boldsymbol{l}_{att_N}]^T \tag{1}$$

where $N$ depends on the number of attribute types which you want to integrate and $\boldsymbol{K}$ is constant through training. The value matrix is initialized as follows:

$$\boldsymbol{v}_{att_j} = \sum_{i \in \{i_1, \ldots, i_M, i_{candidate}\}} \boldsymbol{e}_{att_j}^i = \sum_{i \in \{i_1, \ldots, i_M, i_{candidate}\}} (\boldsymbol{e}_i + \boldsymbol{l}_{att_j}) \tag{2}$$

$$\boldsymbol{V} = [\boldsymbol{v}_{att_1}, \ldots, \boldsymbol{v}_{att_N}]^T \tag{3}$$

There can be many one-to-multiple relations in the KB. For instance, an apple has multiple flavor molecules. Therefore, the entity vector $e_{att}$ should not be an ingredient specific vector and we use $\boldsymbol{e}_i + \boldsymbol{l}_{att}$ instead of using $e_{att}$.

### 4.3 SCORE FUNCTION

Finally, we define our score function as the relationship between the pre-selected ingredient set vector $\boldsymbol{p}$, the candidate ingredient vector $\boldsymbol{q}$, and the relational vector $\boldsymbol{r}$:

$$s(\boldsymbol{p}, \boldsymbol{q}, \boldsymbol{r}) = CosSim(\boldsymbol{p}, \boldsymbol{q}) + CosSim(\boldsymbol{p} + \boldsymbol{q}, \boldsymbol{r}) \tag{4}$$

where $CosSim$ is the cosine similarity, and good pairs will have high scores etc.

Figure 2 shows the geometric differences between our score function and other possible ones. Figure2(a) tries to place the ingredient set and each ingredient into the same spot in vector space, Figure2(b) learns to fit the ingredient set and each ingredient with the relational vector $\boldsymbol{r}$. This score function is often used in the domain of collaborative filtering. The score functions a and b are suggested in previous work. Tay et al. (2018) reported $-||\boldsymbol{p} + \boldsymbol{r} - \boldsymbol{q}||$ could achieve better performance than $-||\boldsymbol{p} - \boldsymbol{q}||$ in the domain of collaborative filtering, which is the user-item based recommender. However, in the food pairing task, the result was the opposite. It seems if we use $-||\boldsymbol{p} + \boldsymbol{r} - \boldsymbol{q}||$, because of $\boldsymbol{r}$, $\boldsymbol{p}$ and $\boldsymbol{q}$ can not be trained properly. As $\boldsymbol{r}$ approaches $\vec{0}$, the performance is improved, so the representation of $\boldsymbol{r}$ cannot be learned. This makes sense, since the ingredient set is a mixture rather than a list of ingredient and ingredient sets can be seen as a single processed ingredient.

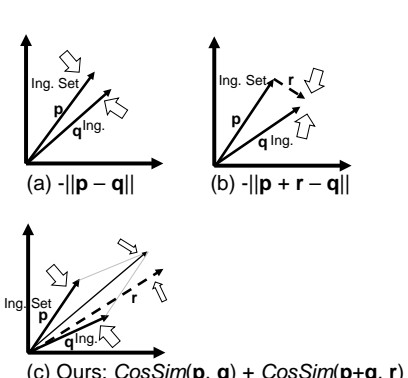

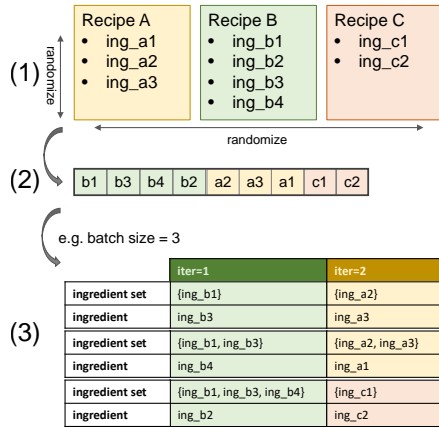

Figure 2: Geometric comparisons of score functions.

Figure 3: How to generate pairs for mini-batches.

Therefore, we propose to use a score function (Figure2(c)) that tries to place the ingredient set $p$ and each ingredient $q$ into the same location in vector space and rotate the relationship between $p$ and $q$ closer to the same place as a relational vector $r$ which is represented using attributes from KB. In addition, since our network structure is symmetric with respect to $p$ and $q$, we require a symmetric bilinear score function .

### 4.4 OBJECTIVE FUNCTION

Our objective function is defined as:

$$L = \sum_{x=1}^{Batch} \sum_{y=1}^{Pos} -log\left[\frac{\exp\left(\frac{s(\boldsymbol{p}_x, \boldsymbol{q}_y, \boldsymbol{r}_{xy}) - \lambda}{\tau}\right)}{\exp\left(\frac{s(\boldsymbol{p}_x, \boldsymbol{q}_y, \boldsymbol{r}_{xy}) - \lambda}{\tau}\right) + \sum_{z=1}^{Neg} \exp\left(\frac{s(\boldsymbol{p}_x, \boldsymbol{q}_y^z, \boldsymbol{r}_{xy})}{\tau}\right)}\right] \qquad (5)$$

where $\lambda$ is the margin that separates the golden pairs and corrupted pairs, $\tau$ is a temperature parameter, $Batch$ is the mini-batch size, $Pos$ is the number of positive examples, $Neg$ is the number of negative examples for each positive example, and the score function for negative examples take the same relational vector as their positive example. In order, to define this objective function, we combine the Batch All loss (Hermans et al. (2017)) with the effective triplet loss in Metric Learning and the additive margin in the softmax loss (Wang et al. (2018)). Note that while the hinge loss is also possible, we found that the softmax loss has better performance and is more stable.

### 4.5 TRAINING

Using pre-processed recipes (removed all ingredients without attributes and empty recipes), we train our models in the following steps (Figure 3): At first, we randomize the order of recipes and its ingredients (Figure 3(1)), and then generate the sequence of ingredients from recipes (Figure 3(2)). After that, we generate pairs between a pre-selected ingredient set and a candidate ingredient from the sequence (see Figure 3/3) and loosely make pairs from same recipes belong to the same mini-batch. Next, we sample additional positive examples as necessary which are taken from other recipes randomly. Finally, we sample negative examples randomly. In the end, mini-batches are generated. Note, we also trained our models without the recipe restriction. However, performance was worse in both one-to-one and many-to-one pairs.

## 5 EVALUATION

We evaluated the the Implicit model and the Explicit model on two tasks against standard baselines and we analyzed the relational representations in terms of interpretability. We propose two tasks for evaluation

For evaluation, we first use the same ranking procedure as in De Clercq et al. (2016), namely the *Recipe Completion Task*. In the Recipe Completion Task, the recommender ranks all ingredients to predict one randomly eliminated ingredient, and the remaining ingredients of each recipe are used as input to the model – which is the pre-selected ingredient set in our definition. We adopt three evaluation metrics for this task (same as the previous work); Mean Rank: the mean position of the eliminated ingredient in the ranked ingredients, Hit Ratio(HR)@10: the proportion of correct ingredients ranked in the top 10, and ROC-AUC(AUC): the mean area under the ROC curve. This task is useful as it allows us to measure the basic performance of our model in the same setting as existing baselines (many-to-one pairing) and against ground truth data.

In order to understand the model performance further, we tested on a second task called *Artificial Food Pairing Task*. In this task, we generate pairs from existing recipes based on ingredient co-occurrences. Given some ingredients as a pre-selected ingredient set and a candidate ingredient, pairs that occur in any of recipes are used as a positive example, otherwise as a negative candidate example. The Artificial Food Pairing task consists of positive pairs and negative pairs where the same pre-selected ingredient set is always used for both positive and negative pairs but positive and negative examples are randomly taken from its candidates with a pre-specified number and ratio. In this task, the recommender predicts whether pairs are positive or negative examples. We use this task for more detailed analysis. We measure MAP@10: the mean average precision in top-10 and AUC as metrics.

We evaluate on two datasets. The first dataset is taken from the CulinaryDB (Bagler (2017)), and the second dataset is from Flavornet (Ahn et al. (2011)). Both datasets contain the recipe set which consist of the list of ingredients and the cuisine categories (e.g. Chinese, Japanese) and the ingredient sets which consist of the names, the food categories (e.g. Meat, Fruit), the flavor molecules the ingredient has. The statistics of datasets shows in Table 4. Before training models, the recipe set are divided into a train, a validation and a test set. On the other hand, we generate triplets (Figure 7) from whole ingredient set in order to train TransE, 172,207 triplets for CulinaryDB and 40,952 triplets for Flavornet.

## 5.1 BASELINES

We compare our Implicit/Explicit models to the following baselines.

- **FREQ**: Frequency predictor that always recommend the most frequently used ingredients of the training recipes. Despite its simplicity it is often a strong baseline in certain domains.
- **PMI**: Recommender based on the PMI score.
- **TFIDF**:Recommender based on the TF-IDF score.
- **NMF** (De Clercq et al. (2016)): Non-negative matrix factorization based recommender. The model is trained using the train and the validation recipes. It is implemented by ourselves.
- **NeuMF** (He et al. (2017)): Neural matrix factorization model which is based on the Neural collaborative filtering framework. We use our Ingredient Embedding layer and Ingredient Set Encoder (Section 4.1) as embedding layers of this model. And the training process is also same as ours (Section 4.5). More details of implementation can be found in section A.5.
- **WIDE&DEEP** (Cheng et al. (2016)): Wide & Deep model based recommender. We use a pre-selected ingredient set, a candidate ingredient, and attributes (Table 4) as inputs of this model. And the training process is same as ours (Section 4.5). More details of implementation can be found in section A.5.

## 5.2 RESULTS AND DISCUSSION

**Predictive performance** Table 1 shows the results on all datasets for all compared baselines in Recipe Completion Task. While both our Implicit and Explicit models outperform all counterparts on all metrics usually with a clear margin, our Explicit model is approximately same performance as the Implicit model. On the other hand, NMF is the best baseline. Performance is good since the Recipe Completion task requires finding only one missing ingredient - even though there should

Table 1: Experimental results on the Recipe Completion Task.

| Datasets | CulinaryDB | | | | Flavornet | | | |
|---|---|---|---|---|---|---|---|---|
| Metrics | Mean Rank | HR@10 | HR@20 | AUC | Mean Rank | HR@10 | HR@20 | AUC |
| FREQ | 473.5 | 0.119 | 0.140 | 0.605 | 257.9 | 0.131 | 0.153 | 0.621 |
| PMI | 612.5 | 0.055 | 0.056 | 0.527 | 346.9 | 0.066 | 0.069 | 0.531 |
| TFIDF | 478.5 | 0.055 | 0.055 | 0.598 | 261.3 | 0.055 | 0.089 | 0.613 |
| NMF | 57.0 | **0.435** | 0.559 | 0.900 | 36.3 | 0.479 | 0.599 | 0.896 |
| NeuMF | 37.5 | 0.400 | 0.530 | **0.943** | 35.7 | 0.332 | 0.509 | 0.898 |
| WIDE&DEEP | 43.5 | 0.350 | 0.478 | 0.929 | 37.7 | 0.351 | 0.493 | 0.896 |
| IRRM(Implicit) | **35.9** | 0.391 | 0.567 | **0.943** | **28.1** | **0.485** | **0.631** | **0.926** |
| IRRM(Explicit) | **35.4** | 0.397 | **0.575** | **0.944** | **28.3** | 0.480 | 0.629 | **0.925** |

Table 2: Experimental results on the Artificial Food Pairing Task (CulinaryDB).

| Pair type | one-to-one | | two-to-one | | three-to-one | |
|---|---|---|---|---|---|---|
| Metrics | MAP@10 | AUC | MAP@10 | AUC | MAP@10 | AUC |
| NMF | 0.638 | 0.702 | 0.604 | 0.786 | 0.601 | 0.794 |
| NeuMF | 0.841 | 0.905 | 0.745 | 0.923 | 0.747 | 0.919 |
| WIDE&DEEP | 0.787 | **0.919** | 0.705 | **0.930** | 0.726 | **0.926** |
| W/O-RELENC | 0.574 | 0.736 | 0.494 | 0.731 | 0.499 | 0.746 |
| IRRM(Implicit) | 0.839 | 0.904 | 0.755 | 0.921 | **0.792** | **0.925** |
| IRRM(Explicit) | **0.842** | 0.906 | **0.759** | 0.922 | **0.790** | **0.926** |

be multiple correct ingredients in the real situation. It appears that our approaches are effective for many-to-one pairs which are constrained by recipes.

In order to evaluate the pairing skills from one-to-one to many-to-one, we use the Artificial Food Pairing Task. We report the results of one-to-one, two-to-one, and three-to-one. We sample positive examples and negative examples according to three ratio settings like $(Pos, Neg) \in \{(20, 80), (50, 50), (80, 20)\}$. Beforehand we classify all ingredients into two categories: $\{Low, High\}$ depending on the number of ingredient occurrences in recipes. Then, pre-selected ingredient sets and positive examples are randomly sampled from each category set to have the same ratio, and negative examples are randomly sampled from all candidates. We adopt this process since generally speaking low frequency ingredients are more difficult to handle than high frequency ingredients. In the end, We prepare 4,536 tasks by pre-selected ingredient set size, where each task consists of 100 pairs.

As can be seen in Table 2, our models outperform NMF on one-to-one, two-to-one, and three-to-one test sets. The proposed models show stable performance as the number of pre-selected ingredients increases.

Here, we only report the results of the Explicit model since the Implicit model shows the same tendency. As expected, L-L seems to be the most difficult task for our models followed by H-L. This result is good since it shows that our model does not overfit to low frequency ingredients.

Since one of the main contributions of the paper is the Relation Encoder Model, We performed an ablation study and evaluated a version of our model without it (W/O RELENC model). W/O RELENC has similar performance to NMF on the one-to-one Artificial Food Pairing task but fails on two-to-one and three-to-one test sets in MAP@10. This clearly show that the Relation Encoder module plays an important role in our models. For more detailed analysis, we compare Roc-curves for our models by ingredient frequency-based pair types(Figure 4), which consists of High Frequency-to-High Frequency(H-H), High Frequency-to-Low Frequency(H-L), Low Frequency-to-High Frequency(L-H), and Low Frequency-to-Low Frequency(L-L), e.g. H-L describes pairs that a pre-selected ingredient is used in various recipes frequently, while the paired ingredient is used in only few recipes.

**Interpretablity** We first analyzed attention weights in the Explicit model for some specific food pairs around chocolate (see Figure 5). The data shows that *egg* is paired with *chocolate* because of correlations in food category. *miso* on the other hand has considerable flavor molecule related affinity to chocolate. This interpretation for *eggs* is consistent with the results reported by De Clercq et al. (2016). Next, we focus on the attention weights for flavor molecules in our trained Explicit model for a quantitative analysis. We compare the average Flavor Molecule attention weights with correlation coefficients which are calculated on full recipes. Details are in the caption of Table 3. In

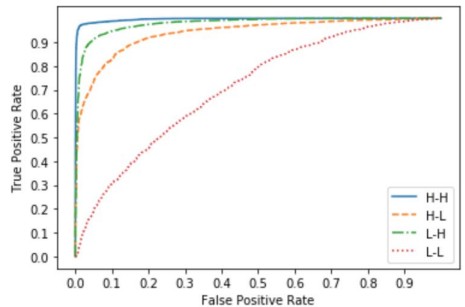
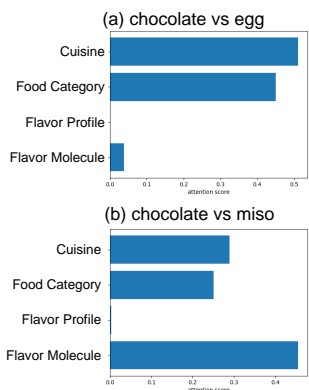

Figure 4: Comparison of Roc-curve by ingredient frequency-based pair type, which use the Explicit model on CulinaryDB, one-to-one pairs. L describes Low frequency, H describes High frequency.

Figure 5: Visualizations of attention weights of our Explicit model on CulidnaryDB.

Table 3: Comparison of food ingredients sorted in descending order of scores. Left: ranked based on the average of Flavor Molecule attention weights in our Explicit model. Right: ranked based on correlation coefficients between the number of food ingredient co-occurrence in recipes and the shared flavor molecules number between two food ingredients. Both scores are calculated on co-occurring ingredient pairs in recipes. Note that while the range of scores for our attentions weights is $[0, 1]$, the range of scores for correlation coefficients is $[-1, 1]$ and Freq is the number of ingredient occurrences in whole recipes.

| Rank | Our attention weights | | | Correlation coefficients | | |
|---|---|---|---|---|---|---|
| | Name | Score | Freq | Name | Score | Freq |
| 1 | soy yogurt | 1.000 | 2 | true frog | 0.730 | 4 |
| 2 | wheaten bread | 1.000 | 4 | florida pompano | 0.577 | 3 |
| 3 | sandalwood | 1.000 | 2 | shellfish | 0.510 | 7 |
| 4 | common dab | 1.000 | 2 | oil palm | 0.488 | 3 |
| 5 | blackberry brandy | 1.000 | 3 | abalone | 0.408 | 4 |
| 6 | miso | 0.773 | 95 | orange oil | 0.403 | 8 |
| 7 | wine | 0.618 | 299 | multigrain bread | 0.358 | 8 |
| 8 | sesame | 0.603 | 1394 | potato bread | 0.356 | 7 |
| 9 | gelatin | 0.583 | 238 | waffle | 0.352 | 13 |
| 10 | sherry | 0.578 | 475 | fruit juice | 0.344 | 21 |

the ranking of attention weights, up to the top 5 are high score ingredients that do not seem to be flavor molecule correlated. It would appear that since the number of ingredient occurrences is very low, the relational representations are not learned properly. On the other hand, the top 6 and below looks like a fairly good result. In the ranking of correlation coefficients, there are some ingredients like *breads* and *waffle* that seems to have little to do with flavor molecules. In consequence the model is able to find hidden ingredients structures within recipes different from simple correlation coefficients.

# 6 CONCLUSION

We proposed interpretable models for food ingredient recommendation systems that rank ingredients based on the co-occurrence of ingredients in cooking recipes and learned relational representations. In our experiments, we found that explicitly and implicitly integrated factors can improve predictions, and our trained relational representations detect interesting correlations between ingredients and factors such as molecular structure. Future work will focus on controlling ranking scores based on manual modifications of relational representations by domain experts (e.g. chefs and food professionals).

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

## A  APPENDIX

### A.1  IMPLEMENTATION DETAILS AND HYPERPARAMETERS

We implemented our models in PyTorch[1] on a Linux machine. For tuning the hyperparameters, we select the model that performs best in validation set based on the precision metric. We report results on the test set. All models are optimized using the Adam optimizer (Kingma & Ba (2015)), and are trained until convergence, the maximum is 35 epochs. The dimensionality $d$ and $d^{KB}$ is tuned amongst $\{64, 128, 200\}$, the batch size $Batch$ is tuned $\{16, 32, 64\}$, the number of positive examples $Pos$ is tuned $\{1, 2, 3, 5, 10\}$, the number of negative examples is tuned $\{1, 2, 3, 5, 10\}$, the learning rate is tuned $\{0.1, 0.01, 0.001\}$, the margin $\lambda$ is tuned $\{0.1, 0.2, 0.25, 0.5\}$, the temperature $\tau$ is tuned $\{0.5, 1, 10\}$. Since CulinaryDB has 4 attribute types and Flavornet has 3 attribute types, we use the memory size $N_{CulinaryDB} = 4$ and $N_{Flavornet} = 3$ respectively in our Implicit/Explicit models.

For most datasets, we found the following hyperparameters work well: $d = d^{KB} = 128$, $Batch = 64$, $Pos = 1$ is for Recipe Completion Task, $Pos = 2$ is for Artificial Food Pairing Task, $Neg = 5$, learning rate $= 0.001$, $\lambda = 0.2$, and $\tau = 1$. Ingredients with less than 1 occurrence in the train set are learned as the UNK ingredient.

---

[1]https://pytorch.org/

## A.2 INGREDIENT SET ENCODER

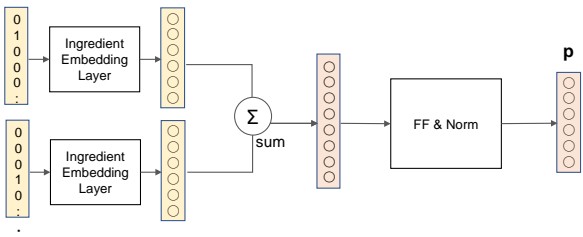

Figure 6: Ingredient Set Encoder

## A.3 TRIPLETS FOR INGREDIENTS

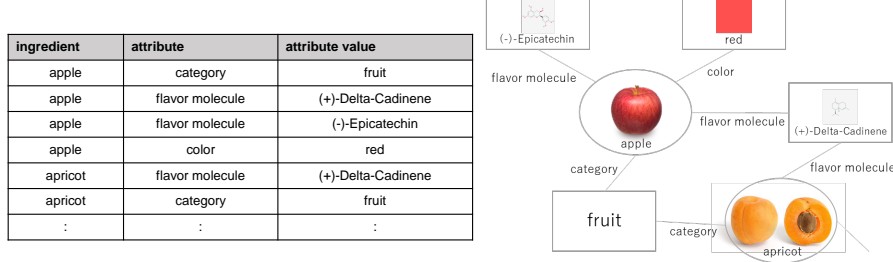

| ingredient | attribute | attribute value |
|---|---|---|
| apple | category | fruit |
| apple | flavor molecule | (+)-Delta-Cadinene |
| apple | flavor molecule | (-)-Epicatechin |
| apple | color | red |
| apricot | flavor molecule | (+)-Delta-Cadinene |
| apricot | category | fruit |
| : | : | : |

Figure 7: Triplets for Ingredients

## A.4 DATASET STATISTICS

Table 4: Dataset Statistics

| Datasets | Recipe Num | Ing./Molecule Num | Attribute Types |
|---|---|---|---|
| CulinaryDB | 45,772 (train+valid: 41,231, test: 4,541) | 658/1,788 | cuisine category, food category, flavor molecules, flavor profile(e.g. Bitter) |
| Flavornet | 55,001 (train+valid: 49,501, test: 5,500) | 381/1,021 | cuisine category, food category, flavor molecules |

## A.5 THE IMPLEMENTATION DETAILS OF NEUMF AND WIDE&DEEP

We implemented both the NeuMF model and the Wide&Deep model from scratch in PyTorch[2].

We trained the NeuMF model without pre-training. The size of predictive factors is 8, then the architecture of the neural CF layers is $32 \rightarrow 16 \rightarrow 8$, and the embedding size is 16.

In the Wide&Deep model, the wide component consists of the cross-product transformation of pre-selected ingredient sets and candidate ingredients. For the deep part of the model, a 16-dimensional

---

[2]https://pytorch.org/

embedding vector is learned for each categorical feature. We concatenate all the embeddings together with the dense features, resulting in a dense vector of approximately 80 dimensions.

## A.6 Example Ranking Results on the CulinaryDB

Table 5: Example Ranking Results on the CulinaryDB

| Rank | Apple | Chocolate | Apple and Chocolate |
|---|---|---|---|
| 1 | canola oil | almond | canola oil |
| 2 | honey | banana | orange |
| 3 | mint | orange | almond |
| 4 | orange | cream | honey |
| 5 | cinnamon | raisin | raisin |
| 6 | white bread | cream cheese | mint |
| 7 | nutmeg | syrup | cream |
| 8 | lime | mint | corn |
| 9 | rice | canola oil | lemon juice |
| 10 | corn | corn | cinnamon |

