# OpenReview forum: "Interpretable Relational Representations for Food Ingredient Recommendation Systems"
_ICLR.cc/2021/Conference — Reject_

### Official Review · AnonReviewer1 · 2020-10-22
**Overall this paper addresses an interesting problem. However, the paper is poorly written and proposed approaches are not novel.**

**Rating:** 3
**Confidence:** 4

**Review:**

Reject.
Summary
This paper works on the problem of create new recipes. It uses recommendation approaches on it. The paper use two approaches for explainability and conducted experiments on two real-world datasets.

Stregthens
1. The explainability in recommendation is an important research problem.
2. The recommendation for food pairing is reasonable.
3. This paper demostrate the proposed approach on two real-world datasets.


Weakness
1. This paper's writing can be greatly improved. It needs proofwriting. There are a lot of typos and also unfinished sentences. (See minors for details)
2. The novelty of this paper is low. Most of the techniques are not proposed in this paper. For example the memory network in already used for explainability in recommendation system. [Huang 2018].
3. The compared approaches are very simple and outdated. There is no related work section of position this paper in the literature. There are many advanced recommendation approaches. The authors can do a better literature Survey. A few example below
- He, Xiangnan, et al. "Neural collaborative filtering." Proceedings of the 26th international conference on world wide web. 2017.
- Cheng, Heng-Tze, et al. "Wide & deep learning for recommender systems." Proceedings of the 1st workshop on deep learning for recommender systems. 2016.
4. The paper mentions "While both our Implicit and Explicit models outperform all counter parts on all metrics usually with a clear margin". However, from figure 1, NMF is the best performing one on HR@10 on CulinaryDB.


Minors:
In introduction line 2: "food preference modelling", -> modeling,
All the "paring" should be "pairing"
Right above Section 2. Missing peorid at the end of the sentence.
"At the embedding layer,this", add space after comma
"In this task we" => In this task, we
"eventhough" => even though
"This result is some good " need to change.

Questions
"where the only concern is whether a user has interacted with an item and the system ..." Why this is a concern? Is it an approach?
For section 3.5, it has a three steps for generating a mini batch.It mentions sample additional postiive examples as necessary. Why sampling addtional positive examples are needed?
"Note that although we also trained our models without the recipe restriction, " What does it mean by without recipe restrictions?
The "Artificial Food Pairing Task" is a bit confusing. Does a pair mean two ingredients or mean one ingredient set and one ingredient?

---

### Official Review · AnonReviewer4 · 2020-10-25
**Interesting Problem but Important Technical Components Missing**

**Rating:** 5
**Confidence:** 5

**Review:**

This paper tackles ingredient recommender systems problem. This paper proposes the Interpretable Relational Representation Model (IRRM) to achieve both usefulness and interpretableness. There are two variants of the model, first is to model latent relation between two ingredients, the second is to leverage external knowledge base and results from TransE to learn relational representations.

The problem setting is interesting: using ingredient recommender systems to help chefs to create better or more creative ingredient combinations. This topic relatively recessives little attention in recommender system community but it seems worth a while pursuing this direction.

I like the way the paper models the problem but there are several technical components are too important to miss:

1. The baseline methods chosen are too simple to be compared against. FREQ, PMI and TFIDF are simple rule-based methods and NMF is the only commonly used method in recommender system but matrix factorization is too weak a baseline given that the proposed method is a neural network approach. This is also related to second point.

2. Connections of this problem to Sequential Recommendation is largely missing. It seems to me this problem is actually closed related to sequential or session-based recommendations. Because the problem can be formulated as given a list of ingredients the recipe has interacted with, what is the next ingredient that this particular recipe is most likely to engage next? Here ingredient is the item and recipe is the user in sequential recommendation, so all the methods that have been developed for sequential recommendation can be used for this problem. Worth noting methods in this category is 2018 SASRec paper (https://arxiv.org/abs/1808.09781) and 2020 SSE-PT paper (https://dl.acm.org/doi/abs/10.1145/3383313.3412258).

3. Many ablation studies are missing. Going over the entire paper, I am not entirely sure what components are most important to the good performance of the model. Many neural network design choices seem too arbitrary to me, e.g. why we need to add p and q in Figure 1. A rigorous approach should be able to convince readers that each choice of model design is the best among all alternatives and without it, the performance would suffer.

4. Regarding qualitative study, I was just wondering if it is possible to ask crowd-sourcing chefs to rate some newly created recipes for unseen recipes. This way the offline results would be convincing because we are not quite sure that better offline ranking results would necessarily lead to better online performance for this particular problem.

---

### Official Review · AnonReviewer3 · 2020-10-28
**The paper is built upon modification to prior work related to memory network-based recommendations. The main contribution aspect of the paper is to apply this work in a completely new domain of food science. Overall the problem the authors are trying to solve is well defined and their approach is solid.**

**Rating:** 7
**Confidence:** 3

**Review:**

Overall, I vote for accepting. The authors propose a novel approach to support chefs with creating/experimenting with new recipes to overcome the challenging combinations of taste/texture etc. that can result from addition of new ingredients. Their idea of adding interpretability to their results from a Food Knowledge Base has a lot of appeal, especially to end-user’s (chef’s in this case). The author’s proposed solution is well supported by the robust and detailed evaluation results. The concerns detailed in the cons section is mainly related to the readability and clarity of the paper.

######################################################################

Pros:
(1) The model architecture for both implicit and explicit use cases in the paper has a strong appeal with solid foundation.
(2) The objective function is well defined and the reasoning for changing the score function from the original paper’s LRML is well reasoned and convincing.
(3) The proposed model’s efficacy is well supported by empirical experimentation on two large food datasets and good comparisons with established baseline methods.
(4) The motivation for adding KB based on ingredient attributes is well reasoned and gives more ability to understand which ingredient attribute contributed to the prediction result.

Cons:
(1) The readability of the paper can be improved. The authors have used a lot of prior work from different authors as a basis for many critical components of their architecture. But the explainability and the clear motivation behind using the related work is lacking. The authors could have given a brief summary of previous work they are using as critical components of their solution.
(2) For the experiments trying to measure interpretability it requires domain knowledge and understanding of Food Science. For example, Table 3 results which are used to illustrate how the explicit model can help with interpretability requires understanding of the training dataset beyond general statistics. Why are the ingredients ranked on attention weights in Table 3 better related to flavor molecules? But it would probably really appeal to someone who has knowledge pertaining to them.

---

### Official Review · AnonReviewer2 · 2020-10-28
**Promising application paper about ingredient recommendation.**

**Rating:** 5
**Confidence:** 4

**Review:**

The paper studies a promising task of interpretable food ingredients recommendation - there has been a growing interest in modeling recipes. The idea of leveraging KG to improve the interpretability/faithfulness of recipe-related ML tasks seems like a contribution to the community. In particular, the author proposes a method to learn pair specific relational representations for one-to-one (i.e. ingredient to ingredient) and many-to-one (ingredient-set to ingredient) food pairing tasks.

Pros:

The task itself is an interesting application; meanwhile, the task is non-trivial as the ingredient pairing is complicated and affected by various factors.

It uses recipes instead of interaction history to recommend complementary ingredients.

It proposes a method based on the memory network. In particular, it first embeds the preselected ingredient set vector and a candidate ingredient vector, then sums them and feeds into the memory network. The output of the memory network is called the relational vector, which is added with the input embedding and then put into a scoring function. The training follows the standard ranking problem which aims to optimize the triplet loss.

KB triples of (ingredient, attribute, attribute value) are further represented as external KB embeddings to augment the memory network.

Conduct a qualitative analysis of the attention weights on the KB embedding and show that IRRM is able to capture some level of relation between ingredients.


Cons:

The experimental section seems to miss important baselines models. Most of the baseline models are non-neural network-based methods. Also, they are mostly based on interaction history. It would be convincing to adding several neural model baselines that use recipe text as training inputs. Otherwise, it's hard to see which component brings improvement.

The author proposes two new evaluation tasks to show the model's performance - both of them fall into a category of ingredients completion. The tasks seem a bit simple and less beneficial in a real scenario. There are more practical and challenging alternatives that could be used for better evaluation. For example, predict the complementary ingredients given the recipe name or recipe steps.

The improvement of IRRM on Hit@10 on the Recipe Completion Task seems marginal (i.e. it even underperforms NMF). Is there any reason?

One other concern is that - most contribution comes from the novel task. It may have a limited scope of the audience at ICLR as it's more like an industrial track application paper, though from the application perspective the paper may be impactful. Other top venues in the field of data mining and recommender systems seem like a better fit.

---

### Decision · Program_Chairs · 2021-01-07
**Final Decision**

**Decision:**

Reject

**Comment:**

There is a broad consensus that this paper explores an interesting and novel problem space. Nonetheless, in their initial assessment, the reviewers pointed to a few limitations of the paper including lack of strong baselines, lack of an ablation study, and weaker results according to the HIT@10 metric.

The authors provided an improved version of their paper as a response. The new paper added two baselines, is better written, and justifies some of the HIT@10 results (basically, the metric is biased for this task).

After discussion, the reviewers find that the contribution of the current manuscript falls short of the acceptance threshold.  In particular, the reviewers find that: 1) this contribution is for a specific domain of recommender systems, an area of interest, but perhaps only relevant to a subset of the ICLR community; 2) while more recent baselines helped, there has been lots of more recent work on collaborative filtering models for recommender systems over the last few years (the Wide&Deep baseline is from 2016); 3) since some of the usual recommender systems' metric does not seem appropriate here, why not suggest new ones (or propose a slightly different evaluation protocol); 4) the proposed model is useful, but somewhat incremental given prior work. All in all, while any of these limitations on their own might not have been sufficient to warrant rejection, I find that their combination does.

Given the interest in this new task, I do strongly encourage the authors to pursue their work. I also find that the qualitative study propsoed by Reviewer 4 could add another interesting angle to this paper (I also imagine that it might not be that easy to carry out).